# Reactivation of Hepatitis B Virus in Lung Cancer Patients Receiving Tyrosine Kinase Inhibitor Treatment

**DOI:** 10.3390/jcm12010231

**Published:** 2022-12-28

**Authors:** Po-Hsin Lee, Yen-Hsiang Huang, Yu-Wei Hsu, Kun-Chieh Chen, Kuo-Hsuan Hsu, Ho Lin, Teng-Yu Lee, Jeng-Sen Tseng, Gee-Chen Chang, Tsung-Ying Yang

**Affiliations:** 1Division of Chest Medicine, Department of Internal Medicine, Taichung Veterans General Hospital, Taichung 407, Taiwan; 2College of Medicine, National Yang Ming Chiao Tung University, Taipei 112, Taiwan; 3Ph.D. Program in Translational Medicine, National Chung Hsing University, Taichung 402, Taiwan; 4Rong Hsing Research Center for Translational Medicine, National Chung Hsing University, Taichung 402, Taiwan; 5Institute of Biomedical Sciences, National Chung Hsing University, Taichung 402, Taiwan; 6Cancer Prevention and Control Center, Taichung Veterans General Hospital, Taichung 407, Taiwan; 7Computer and Communications Center, Taichung Veterans General Hospital, Taichung 407, Taiwan; 8Division of Pulmonary Medicine, Department of Internal Medicine, Chung Shan Medical University Hospital, Taichung 402, Taiwan; 9School of Medicine, Chung Shan Medical University, Taichung 402, Taiwan; 10Institute of Medicine, Chung Shan Medical University, Taichung 402, Taiwan; 11Department of Applied Chemistry, National Chi Nan University, Nantou 545, Taiwan; 12Division of Critical Care and Respiratory Therapy, Department of Internal Medicine, Taichung Veterans General Hospital, Taichung 407, Taiwan; 13Department of Life Sciences, National Chung Hsing University, Taichung 402, Taiwan; 14Division of Gastroenterology and Hepatology, Department of Internal Medicine, Taichung Veterans General Hospital, Taichung 407, Taiwan; 15Department of Post-Baccalaureate Medicine, College of Medicine, National Chung Hsing University, Taichung 407, Taiwan

**Keywords:** hepatitis B virus reactivation, anti-HBc, lung cancer, tyrosine kinase inhibitor

## Abstract

(1) Background: We aimed to evaluate the risk of hepatitis B virus (HBV) reactivation in lung cancer patients treated with tyrosine kinase inhibitor (TKI), particularly in those with resolved HBV infection. (2) Methods: In this retrospective hospital-based cohort study, we screened all lung cancer patients with positive hepatitis B core antibodies (anti-HBc) receiving systemic antineoplastic treatment during the period from January 2011 to December 2020. Cumulative incidences of HBV reactivation, and their hazard ratios (HRs), were evaluated after adjusting patient mortality as a competing risk. (3) Results: Among 1960 anti-HBc-positive patients receiving systemic therapy, 366 were HBsAg-positive and 1594 were HBsAg-negative. In HBsAg-positive patients without prophylactic NUC, 3-year cumulative incidences of HBV reactivation were similar between patients receiving chemotherapy and patients receiving TKI (15.0%, 95% confidence interval (CI): 0–31.2% vs. 21.2%, 95% CI: 10.8–31.7%; *p* = 0.680). Likewise, 3-year cumulative incidences of HBV-related hepatitis were similar between the two groups (chemotherapy vs. TKI: 15.0%, 95% CI: 0–31.2% vs. 9.3%, 95% CI: 2.8–15.7%; *p* = 0.441). In 521 HBsAg-negative TKI users, the 3-year cumulative incidence of HBV reactivation was only 0.6% (95% CI: 0.0–1.9%). From multivariable regression analysis, we found that the only independent risk factor for HBV reactivation in TKI users was HBsAg positivity (HR 53.8, 95% CI: 7.0–412.9; *p* < 0.001). (4) Conclusion: Due to high risks of HBV reactivation in HBsAg-positive TKI users, NUC prophylaxis can be considered. However, in patients with resolved HBV infection, such risks are lower, and therefore regular monitoring is recommended.

## 1. Introduction

Chronic hepatitis B virus (HBV) infection is highly prevalent. It causes a major concern during systemic antineoplastic therapy for cancer patients [1,2]. HBV reactivation may not only delay antineoplastic treatments, but may also cause fulminant hepatitis flare, hepatic failure, and death [3,4]. Among HBV carriers without nucleos(t)ide analogue (NUC) prophylaxis, cytotoxic chemotherapy is a well-known risk factor for HBV reactivation, and the incidence of HBV reactivation can be as high as 20–50% [5,6]. In addition to chemotherapy, tyrosine kinase inhibitors (TKIs) have been reported to increase the risk of HBV reactivation. TKIs such as Bcr-Abl multikinase inhibitors (including imatinib, nilotinib, and dasatinib) are used for chronic myelogenous leukemia, gastrointestinal stromal tumor, and desmoid tumor [7,8]. In lung cancer patients, epidermal growth factor receptor (EGFR)-TKI treatment has been considered as a risk factor for HBV reactivation [9]. Growing evidence suggests the involvement of TKI in HBV reactivation [10].

With repeated reports on HBV reactivation, TKI has been categorized into a class of moderate risk of HBV reactivation in guidelines on clinical practice [11]. According to American Society of Clinical Oncology (ASCO), hepatitis B surface antigen (HBsAg)-positive patients are recommended to start antiviral prophylaxis before TKI therapy [12]. However, few studies have systematically investigated risks of HBV reactivation among TKI users, without estimated incidence rate on the HBV reactivation. Furthermore, HBV reactivation caused by TKI therapy is rarely reported among patients with resolved HBV infection, which is defined as negative HbsAg but positive hepatitis B core antibody (anti-HBc) present in blood [10]. Particularly, the incidence of HBV reactivation in patients with resolved HBV infection remains undetermined. Here, we aimed to systematically investigate HBV reactivation risks in lung cancer patients receiving TKI, including those being HBV carriers or with resolved HBV infection.

## 2. Materials and Methods

### 2.1. Study Design

This was a retrospective cohort study conducted in Taichung Veteran General Hospital, a tertiary referral center in central Taiwan. We screened all patients with newly diagnosed lung cancer during the period from Jan 2011 to December 2020. Patients with a positive anti-HBc in blood were included in this study. We excluded those patients with hepatitis C virus or human immunodeficiency virus co-infection and also those without a complete HBV survey, i.e., anti-HBc in HbsAg-negative cases. Patients were followed up until October 2021. All methods of this study were approved by the Institutional Review Board of Taichung Veterans General Hospital (IRB CF20175B) and performed in accordance with the Declaration of Helsinki and the relevant guidelines and regulations. 

### 2.2. Study Subgroups

Subjects were categorized according to the status of HbsAg, antineoplastic treatments, and NUC prophylaxis (Figure 1). The HbsAg status was divided into HbsAg negative or positive groups. Three types of antineoplastic treatments were: TKI, chemotherapy, and immune checkpoint inhibitor (ICI). Before the start of antineoplastic treatments, patients either received or did not receive the prophylactic NUC. For those patients who received first-line TKI, followed by second-line chemotherapy without NUC prophylaxis, the risk of HBV reactivation was respectively evaluated during the TKI treatment or chemotherapy period.

### 2.3. HBV Reactivation and HBV-Related Hepatitis

In HBsAg-positive patients, HBV reactivation was defined as one of the following conditions: (1) a ≥ 100-fold increase in HBV DNA relative to the baseline level; (2) HBV DNA ≥ 1000 IU/mL in a patient with prior undetectable level; or (3) HBV DNA ≥ 10,000 IU/mL with an unknown baseline level.14 In patients with resolved HBV infection, i.e., positive anti-HBc and negative HBsAg, HBV reactivation was defined as one of the following conditions: (1) detectable HBV DNA; or (2) reappearance of HBsAg (seroconversion) [13]. HBV-related hepatitis after systemic antineoplastic treatment was defined as follows: serum alanine aminotransferase (ALT) of >2 times the baseline level, and >2 times the upper limit of normal levels [14]. Hepatitis induced by causes other than antineoplastic therapies was not considered to be HBV-related hepatitis.

### 2.4. Major Outcome

The main outcome was the occurrence of HBV reactivation or HBV-related hepatitis. The 3-year cumulative incidence was calculated. To assess the cumulative incidence of HBV reactivation, subjects were followed up from the date of systemic antineoplastic treatment until one of the following had occurred: main outcome, end of clinic follow-up, initiating NUC before another course of antineoplastic treatment, or patient mortality.

### 2.5. Statistics Analyses

To compare intergroup differences for categorical and continuous variables, Fisher’s exact test, Pearson’s chi-square test, and Mann–Whitney U test were used. The cumulative incidences of HBV reactivation/HBV-related hepatitis, and the overall survivals were estimated using the Kaplan–Meier method. Inter-group differences were assessed using the stratified log-rank test. Cumulative incidences were determined after adjusting patient mortality as a competing risk factor. Factors associated with HBV reactivation were assessed using the Cox proportional hazard model. The strength of association was presented as the Hazard ratio (HR) and 95% confidence interval (CI). In this study, we used the two-tailed statistical tests, and significance was set at *p* < 0.05. All analyses were performed on the IBM SPSS Statistics package, version 23 (IBM Corporation, Armonk, NY, USA).

## 3. Results

### 3.1. Study Subjects

The algorithm and categories of the study participants we enrolled are illustrated in Figure 1. A total of 1960 patients with a positive anti-HBc were analyzed. Among them, 366 were HBsAg-positive and 1594 were HBsAg-negative. Their baseline characteristics are shown in Table 1. Most patients were middle aged or older, in their late cancer stages, and with histological adenocarcinoma. Half of them were non-smokers. Two thirds of patients received chemotherapy as first-line treatment, while the remaining one third received TKI.

### 3.2. HBV Reactivation under HBV Prophylaxis

Among HBsAg-positive patients, only a small proportion of TKI users (14.6%) received prophylactic NUC, while most patients (92.7%) received prophylactic NUC before chemotherapy. For patients with resolved HBV infection, the use of NUC prophylaxis was rare (0.4%). Entecavir (75.6%) was the most frequently prescribed NUC, followed by tenofovir (10.0%), telbivudine (9.7%), and lamivudine (4.7%). None of the patients under HBV prophylaxis developed HBV reactivation during the course of antineoplastic treatment.

### 3.3. HBV Reactivation without HBV Prophylaxis

Clinical data of patients developing HBV reactivation after systemic antineoplastic treatment are shown in Table 2. Of the 19 patients with HBV reactivation, the peak ALT level exceeded > 10 times of the upper limits in 5 patients. Two patients died of hepatic failure. Among HBsAg-positive patients receiving chemotherapy with HBV reactivation, all of them developed HBV-related hepatitis. Among HBsAg-positive TKI users, 15 developed HBV reactivation, while 8 met the criteria of HBV-related hepatitis. Only one HBsAg-negative TKI user had developed HBV reactivation. All TKI users with HBV reactivation had received EGFR-TKI. No TKI user had liver failure or died after HBV reactivation.

### 3.4. HBsAg-Positive Patients without HBV Prophylaxis

A total of 104 HBsAg-positive patients received the first-line TKI without prophylactic NUC, with 15 developed HBV reactivation. For patients who received TKI followed by chemotherapy without NUC before chemotherapy (N = 13), we categorized them into the chemotherapy group. On the other hand, there were 20 HBsAg-positive patients (7 as first line and 13 with prior TKI use) receiving chemotherapy without prophylactic NUC, with 3 developing HBV reactivation (Figure 1). As shown in Figure 2A, the 3-year cumulative incidences of HBV reactivation in HBsAg-positive patients receiving chemotherapy and TKI were not significantly different (15.0%, 95% CI: 0–31.2% vs. 21.2%, 95% CI: 10.8–31.7%; *p* = 0.680). Moreover, as shown in Figure 2B, the 3-year cumulative incidences of HBV-related hepatitis were also not significantly different (chemotherapy vs. TKI: 15.0%, 95% CI: 0–31.2% vs. 9.3%, 95% CI: 2.8–15.7%; *p* = 0.441).

### 3.5. HBsAg-Negative Patients without HBV Prophylaxis

Among 1594 patients with resolved HBV infection, 521 received TKI as first-line treatment, with only one developed HBV reactivation. That patient had negative anti-HBS, negative HBV viral load before EGFR-TKI treatment. The patient had HBV reactivation with reappearance of HBsAg, markedly elevated HBV viral load (245,000 IU/mL), as well as high ALT level of 529 U/l after 25.9 months of EGFR-TKI treatment (afatinib followed by osimertinib). No patient with resolved HBV infection developed HBV reactivation after chemotherapy or ICI treatment.

Among HBsAg-negative patients, 149 patients were tested for HBV viral load at baseline (136 patients with negative HBV viral load, 13 patients with detectable viral load). Among the 13 patients with detectable viral load, 6 patients received prophylactic NUC before systemic treatment. For the other 7 patients without prophylactic NUC, none of them developed HBV reactivation. The only one patient developing HBV reactivation among HBsAg-negative patients had undetectable HBV viral load at baseline.

### 3.6. TKI Users without HBV Prophylaxis

Of 349 patients who only received TKI, 16 developed HBV reactivation (15 HBsAg-positive, one HBsAg-negative patient). The group characteristics with or without HBV reactivation are shown in Appendix A. They showed no significant inter-group differences in age, gender, smoking status, histological types, and TKI types. In the reactivation group, all patients showed adenocarcinoma histology. In multivariable analysis, HBsAg positivity was the only independent risk factor for HBV reactivation for TKI users (HR 53.8, 95% CI: 7.0–412.9; *p* < 0.001) (Table 3). As shown in Figure 2C, the 3-year cumulative incidence of HBV reactivation in HBsAg-negative patients receiving first-line TKI without prophylactic NUC was merely 0.6% (95% CI: 0.0–1.9%), significantly different from HBsAg-positive patients (*p* < 0.001).

### 3.7. Overall Survival

To evaluate the overall survival difference between TKI treatment patients with and without HBV reactivation, we focused on HBsAg-positive patients. Those receiving chemotherapy after TKI treatment without prophylactic NUC before chemotherapy were not included for analysis. There was no significant difference of overall survival between patients with and without HBV reactivation. Their median survival was 21.9 months (95% CI, 4.5–39.4) for the reactivation group, and 22.8 months (95% CI, 16.0–29.5) for the non-reactivation group (*p* = 0.582) (Appendix A).

## 4. Discussion

Though TKI is a known risk factor for HBV reactivation, the risk of HBV reactivation has not yet been systematically determined in lung cancer patients. In this study, we found that the incidence of HBV reactivation in HBsAg-positive patients receiving TKI treatment was not low, and therefore prophylactic NUC should be considered. Furthermore, we first explored the incidence of HBV reactivation in lung cancer patients with resolved HBV infection, and the incidence of HBV reactivation was extremely low during systemic antineoplastic treatment. Although prophylactic NUC may not be required, regular monitoring of liver function and HBV viral load during the treatment course is suggested.

Our results clearly showed that the 3-year cumulative incidences of HBV reactivation or HBV-related hepatitis in TKI users were not significantly lower than that of patients receiving chemotherapy. In a previous study, the HBV reactivation rate among HBsAg-positive TKI uses was 9.36% (16/171) [9]. However, the cumulative incidence adjusted by competing mortality risk was not precisely calculated in the absence of control patients only receiving chemotherapy. Our present findings provided strong evidence for HBV prophylaxis in this population. However, we found that the rate of NUC prophylaxis in HBsAg-positive TKI users was quite low, when compared with patients receiving chemotherapy (14.6% vs. 92.7%). The low rate of NUC prophylaxis among TKI users was similar to the previous report [15]. The phenomenon may be contributed to the regulations of National Health Insurance reimbursement in Taiwan. Accordingly, the concept of NUC prophylaxis for HBsAg-positive TKI users needs to be promoted in real-world practice.

Our study first revealed that HBsAg was the most important risk factor for HBV reactivation in TKI users, and the 3-year cumulative incidence of HBV reactivation among TKI users with resolved HBV infection was very low. With a very low risk of HBV reactivation, NUC prophylaxis may not be cost-effective. However, we found that once reactivating HBV, patients might suffer from severe hepatitis flare, with a risk of hepatic failure. Therefore, a regular follow-up on HBV reactivation certainly helps to early detect a possibility of severe hepatitis flare. According to the recommendation from ASCO guidelines, patients with resolved HBV infection should be followed with frequent laboratories every 4 weeks including HBsAg, HBV DNA, and serum ALT. For patients who developed HBV reactivation (defined as appearance of HBV DNA at any level in the guideline), the follow-up frequency increased to every 2 weeks. Once the patient experiences active HBV disease (defined as reverse HBsAg seroconversion from HBsAg-negative to HBsAg-positive or an increase in ALT > 2 times the upper limit of normal levels), antiviral therapy should be given [12].

The mechanism of HBV reactivation among TKI users remains unclear. Tyrosine kinase receptor-mediated signaling pathways are crucial for immune activation and the proliferation of lymphocytes. TKIs block these pathways which lead to impaired lymphocyte function and concomitant HBV reactivation. However, further studies are mandated for determining the mechanisms [16,17].

Several limitations of this study are acknowledged. First, because most TKI users did not receive HBV prophylaxis during the study period, the baseline serum HBV viral load was usually not checked before TKI treatments. Therefore, changes of HBV viral loads from the baseline were not fully evaluated. On the other hand, serum HBV viral loads were typically checked when suspecting HBV-related hepatitis, the risk of HBV-related hepatitis, i.e., one of the major outcomes in this study, was likely precisely evaluated. Second, among HBsAg-positive TKI users, the serum HBV viral load was checked only when hepatitis had occurred, or in preparation for a second-line chemotherapy. The incidence of HBV reactivation would likely be underestimated. However, these factors would not change the conclusion of this study in that the HBV reactivation risk in TKI users was not lower than that of patients receiving chemotherapy. Third, some patients received chemotherapy immediately after TKI discontinuation without prophylactic NUC. They could have complicated the interpretation of the cause of HBV reactivation. Further studies to clarify the role of prior TKI treatment in HBV reactivation during the second-line chemotherapy may be interesting; however, HBV prophylaxis before chemotherapy should not be omitted.

## 5. Conclusions

In lung cancer patients receiving TKI treatment, since the risk of HBV reactivation is not low, NUC prophylaxis should be considered. Given such a low HBV reactivation risk in lung cancer patients with resolved HBV infection, regular monitoring is suggested.

## Figures and Tables

**Figure 1 jcm-12-00231-f001:**
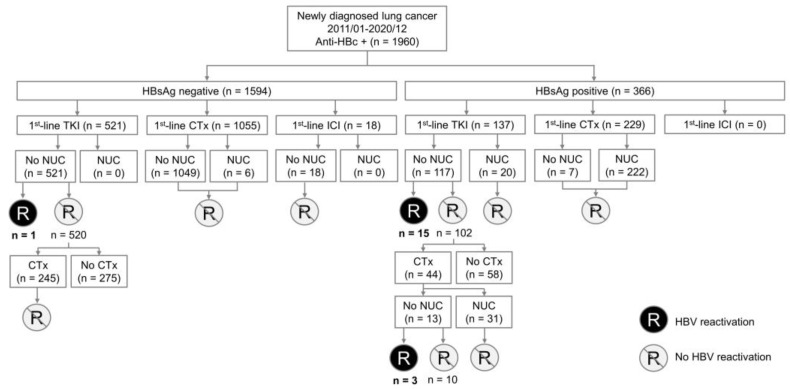
Algorithm for study participants enrollment and the risk of hepatitis B reactivation in various conditions. CTx, chemotherapy; HBV, hepatitis B virus; ICI, immune checkpoint inhibitor; NUC, nuceos(t)ide analogues; TKI, tyrosine kinase inhibitor.

**Figure 2 jcm-12-00231-f002:**
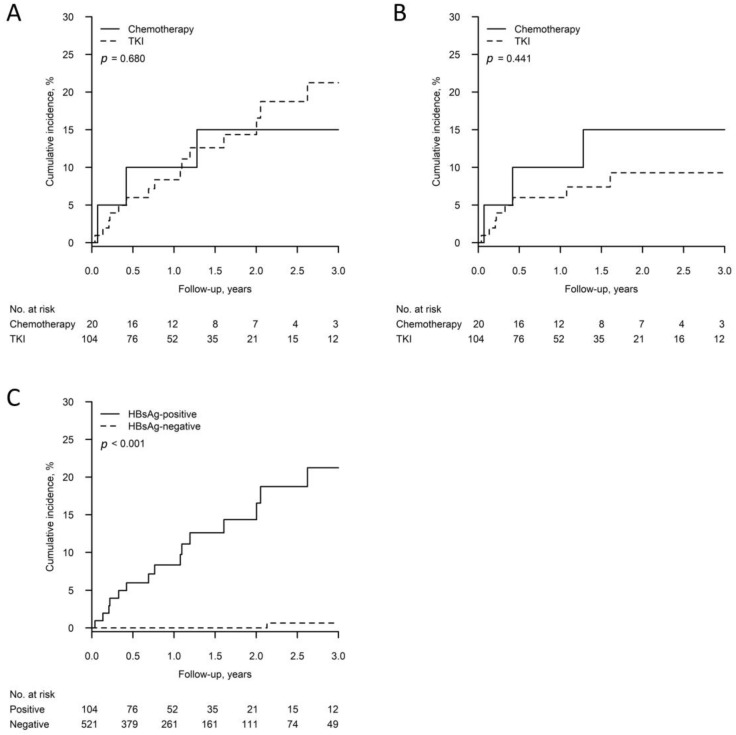
(**A**) Cumulative incidence of HBV reactivation in HBsAg-positive patients without prophylactic NUC who received chemotherapy or tyrosine kinase inhibitor (TKI). (**B**) Comparison of cumulative HBV-related hepatitis incidence between HBsAg-positive patients without prophylactic NUC receiving chemotherapy and TKI. (**C**) Comparison of cumulative HBV reactivation incidence in patients receiving first-line TKI without prophylactic NUC between HBsAg-positive and HBsAg-negative groups. TKI, tyrosine kinase inhibitor.

**Table 1 jcm-12-00231-t001:** Clinical characteristics of the included patients. N/A, not available; TKI, tyrosine kinase inhibitor.

	N = 1960
Age, years (range)	63.0 (55.0–71.0)
Gender, n (%)	
Male	1176 (60.0%)
Female	784 (40.0%)
Smoking, n (%)	
No	955 (48.7%)
Yes	847 (43.2%)
N/A	158 (8.1%)
Stage, n (%)	
I	165 (8.4%)
II	85 (4.3%)
III	349 (17.8%)
IV	1361 (69.4%)
Histology, n (%)	
Squamous cell carcinoma	245 (12.5%)
Adenocarcinoma	1423 (72.6%)
Small cell carcinoma	133 (6.8%)
Others	159 (8.1%)
First-line treatment, n (%)	
Chemotherapy	1284 (65.5%)
Tyrosine kinase inhibitor (TKI) EGFR-TKI Gefitinib Erlotinib Afatinib Osimertinib ALK-inhibitor Crizotinib Ceritinib Alectinib Brigatinib	658 (33.6%) 630 (32.1%) 182 (9.3%) 269 (13.7%) 145 (7.4%) 34 (1.7%) 28 (1.4%) 14 (0.7%) 3 (0.1%) 8 (0.4%) 3 (0.1%)
Immune checkpoint inhibitor	18 (1.0%)
HBV serology, n (%)	
Positive HBsAg	366 (18.7%)
Negative HBsAg and negative anti-HBs	314 (16.0%)
Negative HBsAg and positive anti-HBs	1280 (65.3%)

**Table 2 jcm-12-00231-t002:** Clinical data of patients developing HBV reactivation after systemic anticancer treatment. 3TC, lamivudine; ALT, alanine aminotransferase; ASP8273, an EGFR TKI; ETV, entecavir; LdT, telbivudine; TAF, tenofovir alafenamide; TKI, tyrosine kinase inhibitor. ^#^ The treatment duration in patients receiving first-line TKI only was counted since the initiation of TKI, while those receiving chemotherapy with previous TKI treatment was counted since the initiation of chemotherapy. ^$^ HBV reactivation without HBV-related hepatitis. ^@^ HBV prophylaxis before chemotherapy.

Patient No.	Baseline Characteristics			Antineoplastic Treatment		HBV Reactivation			
	Age/Sex	HBsAg	ALT (U/L)	Medication Courses	Duration ^#^ (Month)	Peak ALT (U/L)	HBV Viral Load (IU/mL)	Rescue NUC Therapy	Clinical Outcome
1	58/F	Positive	48	TKI then chemotherapy	5.1	277	>1.70 × 10^8^	ETV	liver failure
2	67/F	Positive	51	TKI then chemotherapy	0.9	169	1.05 × 10^5^	ETV	ALT decreased
3	60/F	Positive	15	TKI then chemotherapy	15.6	1074	>1.70 × 10^8^	ETV	liver failure
4	45/F	Positive	63	Erlotinib	25.0	19 ^$^	2.25 × 10^4^	ETV ^@^	
5	62/F	Positive	30	Gefitinib	1.9	140	7.03 × 10^3^	ETV	ALT decreased
6	48/F	Positive	20	Gefitinib then osimertinib	19.6	161	8.00 × 10^4^	ETV	ALT decreased
7	50/M	Positive	27	Erlotinib then osimertinib	24.4	16 ^$^	>1.70 × 10^8^	TAF ^@^	
8	57/M	Positive	20	Gefitinib	9.3	65 ^$^	6.44 × 10^7^	ETV	ALT decreased
9	70/M	Positive	19	Gefitinib	1.6	125	2.82 × 10^5^	ETV	ALT decreased
10	57/F	Positive	30	Gefitinib	14.5	31 ^$^	4.51 × 10^4^	ETV ^@^	
11	70/M	Positive	21	Erlotinib	4.0	355	1.03 × 10^5^	LdT	ALT decreased
12	49/F	Positive	12	Erlotinib	13.3	52 ^$^	5.51 × 10^4^	nil	
13	66/F	Positive	20	Afatinib	0.5	1293	>1.70 × 10^8^	ETV	ALT decreased
14	67/F	Positive	32	ASP8273 then gefitinib	5.1	1227	1.11 × 10^5^	ETV	ALT decreased
15	67/M	Positive	40	Afatinib	31.9	58 ^$^	2.47 × 10^6^	nil	
16	60/M	Positive	104	Erlotinib	2.5	191	5.93 × 10^6^	ETV	ALT decreased
17	59/F	Positive	21	Erlotinib then osimertinib	13.1	552	7.93 × 10^6^	ETV	ALT decreased
18	59/M	Positive	44	Erlotinib	8.4	26 ^$^	3.87 × 10^6^	ETV ^@^	
19	67/M	Negative	16	Afatinib then osimertinib	25.9	529	2.45 × 10^5^	ETV + 3TC	ALT decreased

**Table 3 jcm-12-00231-t003:** Univariate and multivariable analysis for HBV reactivation in lung cancer patients receiving tyrosine kinase inhibitor treatment only.

Variable	Total	Univariate Analysis		Multivariable Analysis	
	n = 349	Hazard Ratio (95% CI)	*p* Value	Hazard Ratio (95% CI)	*p* Value
Age		0.96 (0.92–1.00)	0.067	0.97 (0.93–1.01)	0.164
Gender					
Female	204	Reference		Reference	
Male	145	1.63 (0.61–4.35)	0.327	1.18 (0.35–3.97)	0.792
Smoking habit					
Never smoker	244	Reference		Reference	
Ever smoker	75	2.04 (0.74–5.62)	0.167	2.73 (0.72–10.32)	0.140
Tyrosine kinase inhibitor					
Gefitinib	85	Reference		Reference	
Erlotinib	141	0.52 (0.17–1.62)	0.262	0.32 (0.10–1.04)	0.059
Afatinib	77	0.43 (0.11–1.74)	0.239	0.33 (0.08–1.42)	0.135
Osimertinib	23	0.56 (0.07–4.70)	0.596	0.35 (0.04–3.42)	0.365
HBV serology					
Negative HBsAg	276	Reference		Reference	
Positive HBsAg	73	62.20 (8.21–471.08)	<0.001	53.77 (7.00–412.90)	<0.001

## Data Availability

The data presented in this study are available on request from the corresponding author.

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
