# Peer review of "Reactivation of Hepatitis B Virus in Lung Cancer Patients Receiving Tyrosine Kinase Inhibitor Treatment"

_jcm, 2022, doi:10.3390/jcm12010231_

Round 1
Reviewer 1 Report
The paper by Lee et al. analyzes the risk of HBV reactivation in HBsAg-positive or -negative/antiHBc- positive lung cancer patients treated with chemiotherapy (CT) and/or tyrosine kinase inhibitor (TKI).
This study confirms a high risk of reactivation in HBsAg-positive patients, without a significant difference between CT or TKI and a very low risk in HBsAg-negative and antiHBc -positive subjects. These results justify an universal prophylaxis with NUC(s) in overt carriers, but not in those with resolved (past) HBV infection, in accordance with many international guidelines.
The paper is of interest, particularly for the number of patients analysed. However some criticism can be addressed to the authors.
1. Figure 1, compared with Figure 2: the number of HBsAg-positive patients at risk without prophylactic NUC are different--> 117 (TKI) + 7 (CT)= 124 in the Figure 1, 124 in the Figure 2 (A and B), but with a different distribution (104 TKI and 20 CT). The same number of HBsAg-positive treated with TKI without NUC are reported (104 and not 117) in the C section. Please define.
2. Figure 2: the B section descibes the constant HBV-related hepatitis in presence of HBV -reactivation in the patients treated with CT, but not in the subjects treated with TKI who experienced HBV reactivation (hepatitis in only 50% of them). This observation is of interest, in my opinion, and should be described in the text and commented.
3. What about anti-HDV antibodies, particularly in HBsAg-positive patients?
4. Were HBsAg-negative/antiHBc-positive patients tested for HBV DNA at baseline (particularly was HBV DNA at baseline available in the only subject of this group who experienced HBV reactivation; finally, because he was treated with a combination of ETV + 3TC at the time of reactivation)?
5. The authors describe a high risk of HBV reactivation with TKI but a low rate of NUC prophylaxis among these patients was described in the study, as in a previous experience (15). They shoud justify and comment this aspect, apparently contradictory.
6. In the reactivation group all patients showed adenocarcinoma histology. Could the kind of lung cancer influence the rate of HBV reactivation, particullarly in HBsAg-positive patients? Have the authors data regarding other lung cancers (for example microcytoma)?
7. The authors should define in the discussion the strategy of monitoring in low-risk HBsAg-negative/antiHBc-positive lung cancer patients (timing and marker: HBV DNA or HBsAg, as recently proposed by ASCO guidelines?)
Reviewer 2 Report
In the manuscript, Lee et al. screened all lung cancer patients with positive hepatitis B core antibodies (anti-HBc) receiving systemic antineoplastic treatment to evaluate cumulative incidences of HBV reactivation, and hazard ratios. The data demonstrated that HBsAg-positive TKI users are prone to HBV reactivation. In general, the results are well presented and support the conclusions. I believed that the manuscript is interesting. There are several papers describing the correlation between TKI treatment and HBV reactivation. Thus, it is easy to infer that TKI might pose higher risk for HBV reactivation regardless of cancer types. Please highlight the significance of lung cancer in HBV reactivation. Furthermore, it is interesting to discuss the mechanism underlying the induction of HBV reactivation by TKI in this study.
Author Response
In the manuscript, Lee et al. screened all lung cancer patients with positive hepatitis B core antibodies (anti-HBc) receiving systemic antineoplastic treatment to evaluate cumulative incidences of HBV reactivation, and hazard ratios. The data demonstrated that HBsAg-positive TKI users are prone to HBV reactivation. In general, the results are well presented and support the conclusions. I believed that the manuscript is interesting. There are several papers describing the correlation between TKI treatment and HBV reactivation. Thus, it is easy to infer that TKI might pose higher risk for HBV reactivation regardless of cancer types. Please highlight the significance of lung cancer in HBV reactivation. Furthermore, it is interesting to discuss the mechanism underlying the induction of HBV reactivation by TKI in this study.
Response:
Thanks for your suggestion. As you mentioned, there are several studies describing the correlation between TKI treatment and HBV reactivation. However, most studies are case reports and the incidences of HBV reactivation were not clearly defined. The significance of lung cancer in HBV reactivation compared with other malignancy warrants further investigation.
The mechanism of HBV reactivation among TKI users remained unclear. Tyrosine kinase receptor-mediated signaling pathways are crucial for immune activation and the proliferation of lymphocytes. TKIs block these pathways which lead to impaired lymphocyte function and concomitant HBV reactivation. However, further studies are mandated for determining the mechanisms. We added the description in discussion part. (line 267-271)
Reference:
[16] Wang B, Mufti G, Agarwal K. Reactivation of hepatitis B virus infection in patients with hematologic disorders. Haematologica 2019;104:435–43. https://doi.org/10.3324/haematol.2018.210252
[17] Lee P-H, Lee T-Y, Chang G-C, Hepatitis B flare during osimertinib targeted therapy in a lung cancer patient with a resolved hepatitis B virus infection. Eur J Cancer 2020,130:272-4. https://doi.org/10.1016/j.ejca.2020.02.026